# Identifying Spurious Correlations using Counterfactual Alignment

**Joseph Paul Cohen**[†]                                                                joseph@josephpcohen.com
*Stanford University*

**Louis Blankemeier**
*Stanford University*

**Akshay Chaudhari**
*Stanford University*

**Reviewed on OpenReview:** *https: // openreview. net/ forum? id=Utjw2z1ale*

## Abstract

Models driven by spurious correlations often yield poor generalization performance. We propose the counterfactual (CF) alignment method to detect and quantify spurious correlations of black box classifiers. Our methodology is based on counterfactual images generated with respect to one classifier being input into other classifiers to see if they also induce changes in the outputs of these classifiers. The relationship between these responses can be quantified and used to identify specific instances where a spurious correlation exists. This is validated by observing intuitive trends in face-attribute and waterbird classifiers, as well as by fabricating spurious correlations and detecting their presence, both visually and quantitatively. Furthermore, utilizing the CF alignment method, we demonstrate that we can evaluate robust optimization methods (GroupDRO, JTT, and FLAC) by detecting a reduction in spurious correlations.

## 1 Introduction

Challenges related to neural network generalization and fairness often arise due to covariate shift (Moreno-Torres et al., 2012) and shortcut learning (Ross et al., 2017; Geirhos et al., 2020). Shortcut learning can lead to models making decisions based on factors not aligned with expectations of the human that created the model. These powerful models (vision models in this work) leverage a wide array of features and relationships, which may inadvertently incorporate unwanted spurious correlations. These spurious relationships may stem from sample bias (e.g., predicting cows when cows are observed on grass but not on a beach) or may be inherent to the class definition (e.g., predicting cows when an animal has four legs) (Beery et al., 2018).

In this work our objective is to understand black-box classifiers, without access to their training data, as this is a common use case encountered by practitioners. In this analysis, we utilize counterfactual (CF) images which are synthetic images simulating a change in the class label of an image (Wachter et al., 2017). These synthetic images have features modified such that the prediction of the classifier changes. We can then view the synthetic images to understand the reasons that a prediction was made.

Specifically, we are interested in CF images that are directly generated using the gradients of a classifier (Cohen et al., 2021; Joshi et al., 2018; Balasubramanian et al., 2020). Generating CF images with respect to a classifier is rooted in similar logic to that of crafting adversarial examples. However, the key distinction lies in the constraint that CF images remain within the data manifold of plausible images. The latent space of an autoencoder provides such data manifold. This approach enables us to study the specific features used

---

[†]Work not related to position at Amazon.

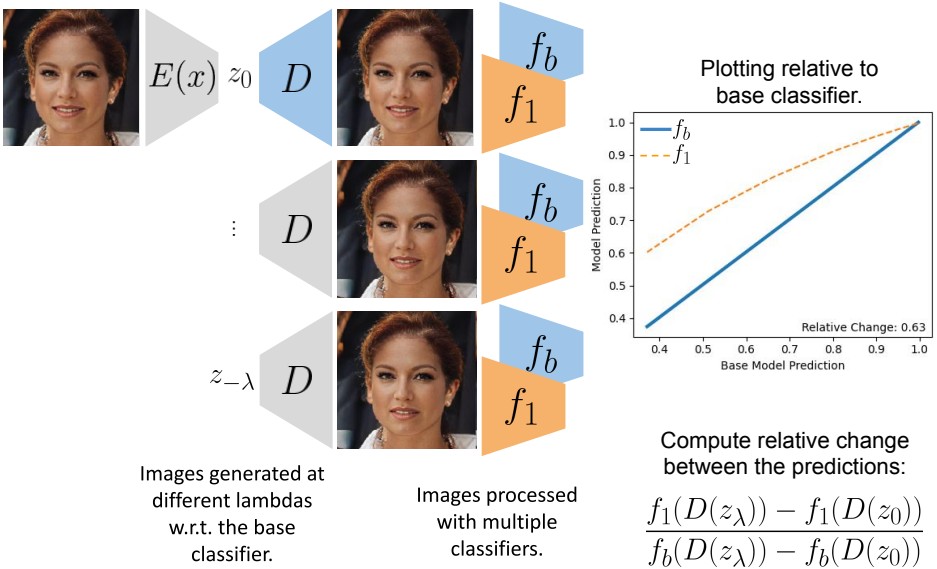

Figure 1: Overview of the alignment methodology. An image is encoded, reconstructed, and then processed by a classifier. The counterfactual is generated by subtracting the gradient of the classifier output w.r.t. the latent representation. The resulting representation is reconstructed back into an image. The reconstructed images are processed with multiple classifiers and the classifier outputs can be plotted side by side to study their alignment. The base model value can be used as the x-axis to more easily compare it to the predictions of another classifier. The output changes can be quantified and compared using relative change.

by a given classifier on a particular input. By keeping the classifier and autoencoder independent, different classifiers can be analysed using a single autoencoder as a reference point.

The task of interpreting these CF images presents new challenges of scale. Spurious correlations may only exist in a handful of samples where rare features occur together. Locating these samples requires investigating the features used for each prediction, which can be automated using the approaches we present in this study.

The approach we take is to generate CF images for the positive predictions of a classifier over an entire dataset and study the resulting CF images using a collection of different classifiers, which we refer to as downstream classifiers, to observe what outputs are impacted. If the CF images generated for one classifier also change the predictions of other classifiers, we can conclude that there is shared feature usage. This shared feature usage can prompt us to investigate if this relationship is unexpected. Such as if a classifier that predicts big noses should share features with a classifier that predicts arched eyebrows.

The relative change metric (Eq. 3) can be used to quantify the impact of counterfactual perturbations and identify spurious correlations that may occur for a specific example or to identify trends across an entire dataset. We study face-attribute classification, recognizing its advantage in allowing visual verification by readers.

Overall, given the problem setting of spurious correlations, the contributions of this work include:

- We propose CF alignment to reason about the feature usage relationships between classifiers, quantified using the relative change metric. This approach allows for both aggregate quantification and targeted querying of models to locate specific examples where the spurious correlations are used to make predictions.

- Our work demonstrates the ability to detect spurious correlations on existing face-attribute and waterbird classifiers. This is validated by observing intuitive trends in a face-attribute classifier as well as inducing spurious correlations and then detecting their presence, both visually and quantitatively.

- We demonstrate that CF Alignment can evaluate robust optimization methods (GroupDRO, JTT, and FLAC) by detecting a reduction in spurious correlations. We also observe improved generalization performance when a spurious correlation is reduced.

## 2 Related Work

Counterfactual (CF) generation can be done in a variety of ways (Verma et al., 2020). Classifier specific approaches generally perturb a latent space representation guided by a classifier. Some methods use the gradient of a classifier to guide movement in the latent space by computing the gradient directly (Cohen et al., 2021) or defining a loss that is optimized (Joshi et al., 2018; Balasubramanian et al., 2020). Another classifier based approach is to train a model that predicts, using training examples, where to walk in the latent space to change the classifier's prediction (Schutte et al., 2020). This has the benefit of working with classifiers that are not differentiable. Another approach is conditional generation which doesn't use a classifier and instead generates images based on a conditioning variable. This conditioning can be a label (Mirza & Osindero, 2014; Baumgartner et al., 2018; Samangouei et al., 2018; Schutte et al., 2020; Hasenstab et al., 2023; Barredo-Arrieta & Del Ser, 2020; Singla et al., 2023) or text (Chambon et al., 2022), or can be provided by manually adjusting the latent representation (Seah et al., 2019). Conditional generation learns a representation from the data and does not capture the exact features utilized by a classifier, as the gradient based methods do.

Several methods exist for studying neural network based classifiers to understand the features they use. Early approaches include gradient based attribute maps (aka saliency heatmaps) that could be overlaid on the image (Simonyan et al., 2014). This was extended to capture more caveats of neural network reasoning (Sundararajan et al., 2017; Springenberg et al., 2015) but these methods still only generate heatmap explanations. Other work focuses on leveraging occlusions to identify discrete image regions (Ribeiro et al., 2016) that are relevant to a prediction. This is useful when predictions are based on discrete features that are smaller than the occlusion size. Other work inspects individual neural network neurons to identify where class information is propagating using linear probing (Alain & Bengio, 2016) and manual identification (Olah et al., 2020; 2017). This is useful for understanding the internal representations of neural networks but does not explain individual predictions.

Along these lines, work by Kim et al. (2018) focuses on concepts represented by layer activations forming so called "Concept Activation Vectors". These vectors allow us to identify inputs which result in a similar internal state and therefore have a similar reason for being predicted. A limitation of this approach is that it is difficult to compare vectors between classifiers because they have different weights.

An approach by Balakrishnan et al. (2021) generates synthetic images using human annotators such that the images have known feature similarity (e.g. a face with all features held constant except for skin color). These images can then be used to identify spurious correlations between attributes. To contrast this approach to ours, this work is creating CFs manually using humans and then looking for unexpected changes in classifier predictions, similar to our approach. A limitation of this method is the required manual effort and the reliance on human knowledge of concepts. The automatic generation of CFs in our work overcomes this issue and allows us to identify features that are unknown initially to humans. And the relative change metric makes it easy to identify potential spurious correlations.

## 3 CF Alignment Methodology

In this work, we introduce the CF alignment approach, outlined in Figure 1. The CF alignment approach tests if a base classifier $f_b$ utilizes features that are also used by a downstream classifier (e.g. $f_1$). If they do, the generated CF would have features modulated which would cause the other classifier (e.g. $f_1$) to have a different output. This alignment can then be measured quantitatively by comparing how aligned the changes in prediction are. Here we consider a classifier as predicting a single scalar value $f(x) \in [0, 1]$. A multiclass classifier can also work if we consider one class vs all other classes to create a single output value.

We say that two classifiers are aligned if both their predictions change negatively with a counterfactual generated for one of them, implying the features removed were used for both predictions. We say they are

inverse aligned when one classifier's prediction is reduced and the other's prediction increased. We discuss how to quantify this in §3.1 but will continue on how we generate CFs.

To generate CF samples a technique known as Latent Shift (Cohen et al., 2021) is used. The implementation of Latent Shift requires the integration of an encoder/decoder model, denoted as $D(E(x))$, where $E$ represents the encoder and $D$ is the decoder. Additionally, a classifier $f$ is incorporated, responsible for predicting the target variable $y$, expressed as $y = f(x)$. It is important to note that both the autoencoder and the classifier are trained independently, with the only specified requirements being differentiability and operation on the same data domain.

To compute a counterfactual, the process begins with encoding an input image $x$ using the encoder $E(x)$, resulting in a latent representation $z$. The next step involves perturbing the latent space to generate counterfactual samples. This perturbation is performed using a base classifier $f_b$, as illustrated in Equation 1. The resulting perturbed latent representation is denoted as $z_\lambda$. Subsequently, the decoder $D$ is employed to reconstruct the image, resulting in a counterfactual image $x'_\lambda$, as depicted in Equation 2.

The perturbation of the latent space, represented by $z_\lambda$, is computed by subtracting $\lambda$ times the gradient of the base classifier $f_b$ with respect to the latent representation $z$. This operation is conducted to induce changes in the latent space that influence the predictions of the classifier. The parameter $\lambda$ is determined through an iterative search process, where its value is systematically adjusted in steps. The objective is to find a suitable $\lambda$ such that the classifier's prediction is either reduced by 0.6 or starts to increase. 0.6 is chosen as a difference that should cross the decision boundary. Just crossing the 0.5 mark is not always large enough to generate a reasonable counterfactual.

$$z_\lambda = z_0 - \lambda \frac{\partial f_b(D(z_0))}{\partial z} \qquad (1) \qquad\qquad x'_\lambda = D(z_\lambda) \qquad (2)$$

Equation 1 expresses the computation of $z_\lambda$, and Equation 2 outlines the generation of the counterfactual image $x'_\lambda$ by decoding the perturbed latent representation.

This process allows us to systematically explore and manipulate the latent space to generate counterfactual samples that reveal insights into the decision-making process of the classifier and its sensitivity to changes in the input data.

### 3.1 Relative Change

Having generated counterfactual samples using the Latent Shift approach, we proceed to algorithmically assess the impact of these samples on various downstream classifiers responsible for predicting the probability of different attributes, denoted as $f_1, f_2$, and so on. These classifiers are distinct from the base classifier, denoted as $f_b$, which was utilized in the counterfactual generation process.

To quantify the relationship between the base classifier and each downstream classifier, we employ the "relative change metric", as defined in Equation 3. This metric is akin to correlation but takes into account not only the direction of change in predictions but also the magnitude of that change. The formula for relative change is expressed as follows:

$$\text{RelativeChange}(f_1, f_b, z_0) = \frac{f_1(D(z_\lambda)) - f_1(D(z_0))}{f_b(D(z_\lambda)) - f_b(D(z_0))} \qquad (3)$$

Here, $D(z_\lambda)$ and $D(z_0)$ represent the reconstructions of the latent representations $z_\lambda$ and $z_0$ (perturbed and original, respectively) by the decoder $D$. The numerator captures the change in prediction made by the downstream classifier $f_1$ in response to the counterfactual perturbation, while the denominator corresponds to the change in prediction made by the base classifier $f_b$ due to the same perturbation.

In our experiments, we opted for relative change over traditional correlation measures. This decision was motivated by the observation that correlation metrics occasionally yielded false positives when only a slight change in the prediction of $f_1$ occurred compared to the base classifier $f_b$. Relative change provides a more nuanced understanding by considering not just the direction but also the magnitude of the change, offering a more robust assessment of the impact of counterfactual perturbations on downstream classifiers. The relative change metric value can be interpreted as follows, a value of 1 indicates a very high alignment, 0 indicates

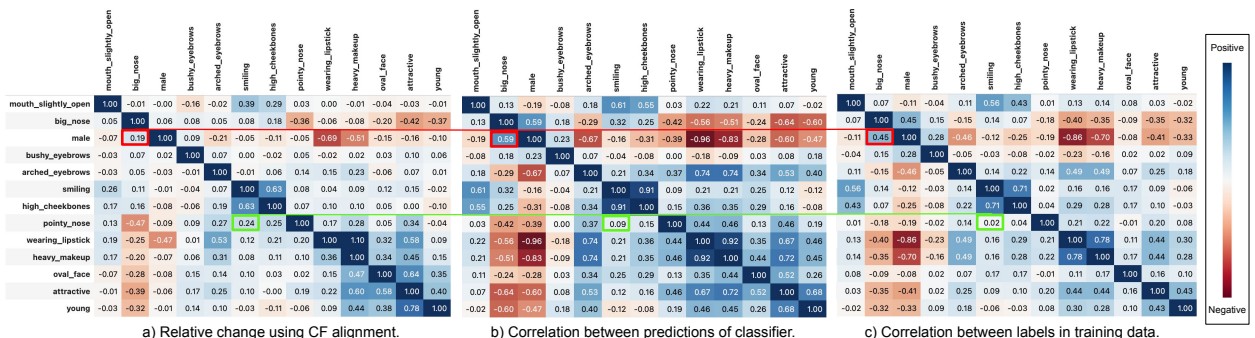

Figure 2: Relationships between face-attribute classifiers as measured by CF-alignment relative changes (left), classifier predictions (middle) and training data labels (right). In (a), base classifiers are along the rows and downstream classifiers are along the columns. Comparing (a) to (b) and (c), shows that many relationships reflected in the CF outputs are preserved from correlations in the training data. We draw the readers attention to some unique differences. The relationship between *male* and *big_nose*, highlighted in red, is strong in both the classifier predictions and ground truth labels but low in CF alignment, indicating that although correlated, these features are not exploited by the classifier. In contrast to this, the relationship between *pointy_nose* and *smiling*, highlighted in green, is weak in both the classifier predictions and ground truth labels but high in CF alignment, indicating that this relationship was introduced by the classifier.

no alignment/independence, and -1 indicates a strong inverse alignment. The relative change metric is a key contribution that allows us to make sense of the CF alignment results at scale over large datasets.

## 4    Experiments

The experiments in this work are performed on the CelebA HQ dataset (Karras et al., 2018) that contains over 200k celebrity images with 40 facial attribute labels per image. The resolution of the images is 1024×1024. Experiments are also performed on the lower resolution (178 x 178) CelebA dataset (Liu et al., 2015).

The pre-trained face classifiers used in this work are sourced from (Vandenhende et al., 2020). They were trained on the CelebA dataset (Liu et al., 2015) to predict 40 different facial attributes on images of dimensions 224×224. These classifiers all have the same architecture, the only difference between them is the last classifier layer weights. We chose this model because it has good performance, is publicly available, and was implemented in PyTorch.

We leverage the VQ-GAN autoencoder from (Esser et al., 2021) trained on the FacesHQ dataset, which combines the CelebA HQ dataset (Karras et al., 2018) and the Flickr-Faces-HQ (FFHQ) dataset (Karras et al., 2019). The resolution of this model is 256x256. In order to make this work for existing classifiers, each classifier normalizes the input image dynamically to match its training domain.

The Captum library (Kokhlikyan et al., 2020) is utilized for baseline attribution methods. PyTorch (Paszke et al., 2019) is used for efficient tensor computation and automatic differentiation. The source code and model weights for all experiments will be released publicly online*. The CF alignment algorithm requires a few seconds to run for each image on a NVIDIA V100 16GB GPU. The CF generation step is the most expensive and is variable due to the automated search for the optimal lambda. The resulting CF images are then processed by each classifier which scales by the number of classifiers studied.

### 4.1    Aggregate statistics over a dataset

Viewing aggregate CF alignment statistics over a dataset can be useful when investigating a model for bias or spurious correlations. To achieve this we compute the average relative change between pairs of classifiers (N=400 images per class) is shown in Figure 2a. Figures 2b and 2c show correlations between classifier

---

*https://github.com/ieee8023/latentshift

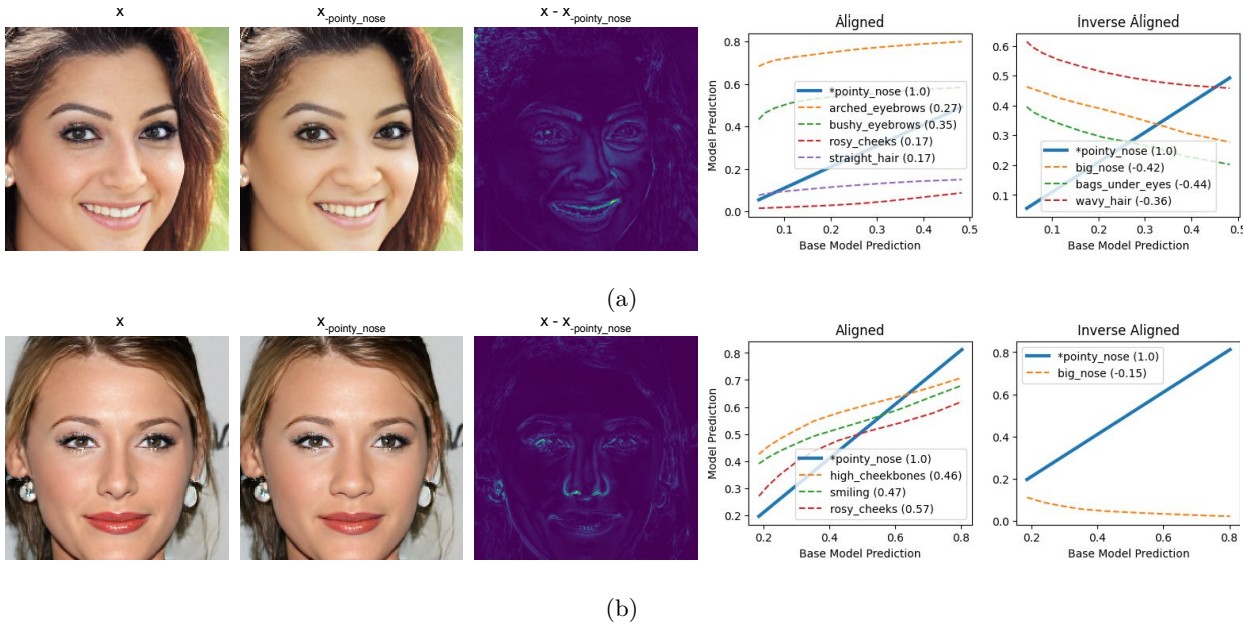

Figure 3: CF alignment examples for the $f_b$=*pointy_nose* with the highest aligned and inverse aligned classifiers. We observe an inverse alignment with *big_nose* and potential spurious relationships with eyebrows, eyes, hair, and smiling. The relative change is shown next to each classifier name.

predictions and the ground truth training labels. A subset of classifiers studied are shown in this figure for clarity, with the full CF alignment matrix in Appendix Fig. 6. These classifiers were chosen because they have intuitive and unintuitive relationships with large relative change that help to illustrate the contribution of this method. The positive relationships between *smiling* and *mouth_slightly_open* and *high_cheekbones* seems intuitive while the positive relationship between *pointy_nose* and *arched_eyebrows* seems unintuitive and likely due to a spurious correlations. This relationship is explored on a single example in §4.2.

The caption of Figure 2 details observations where correlation in the training data and classifier outputs are not the same as the relationships uncovered using CF alignment. These observations indicate that even if correlated attributes are presented to the classifier it does not cause feature usage to be correlated. These observations also imply that even if uncorrelated attributes are presented to the classifier it may still construct spurious feature relationships.

## 4.2 Studying specific examples

Using the aggregate analysis from §4.1 as a guide, classifiers can be selected to investigate undesired relationships on specific images. Inspecting the *pointy_nose* classifier in Figure 3 using two images with positive predictions for *pointy_nose* we observe similar and contrasting relationships between classifiers. A common theme is the inverse alignment with *big_nose* which is intuitive as it is the opposite of a pointy nose. *rosy_cheeks* is a common aligned classifier which does not appear to have an intuitive reason (to the authors) and is likely a spurious correlation. An alignment with *smiling* is only observed in one of the examples.

A takeaway from these examples is the uniqueness of a spurious correlations to specific images with specific features, thus demonstrating the need for the method we present to mine for examples where these spurious correlations can be observed.

## 4.3 Validation by inducing spurious correlations

In order to further verify the CF alignment approach, we construct a classifier with a known spurious correlation and then demonstrate that this bias is observable in the CF alignment plot. A spurious correlation

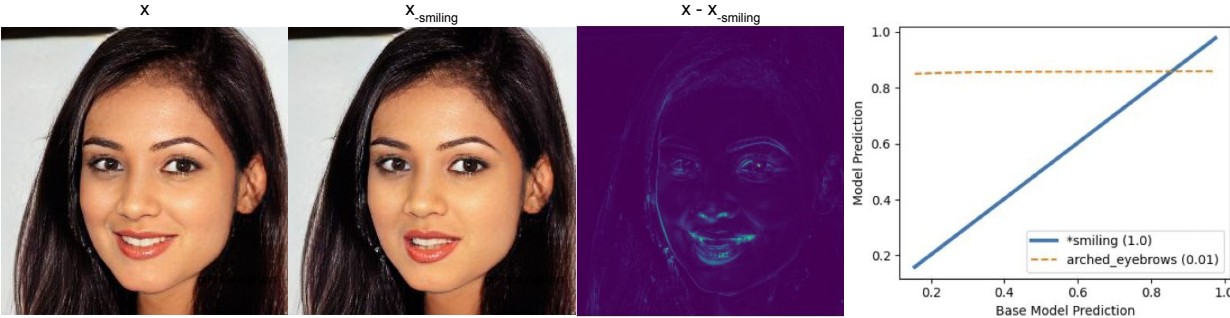

(a) CF for the base classifier *smiling* showing eyebrows are unchanged. The horizontal line indicates the prediction of *arched_eyebrows* is not influenced by the features used for *smiling* in this image.

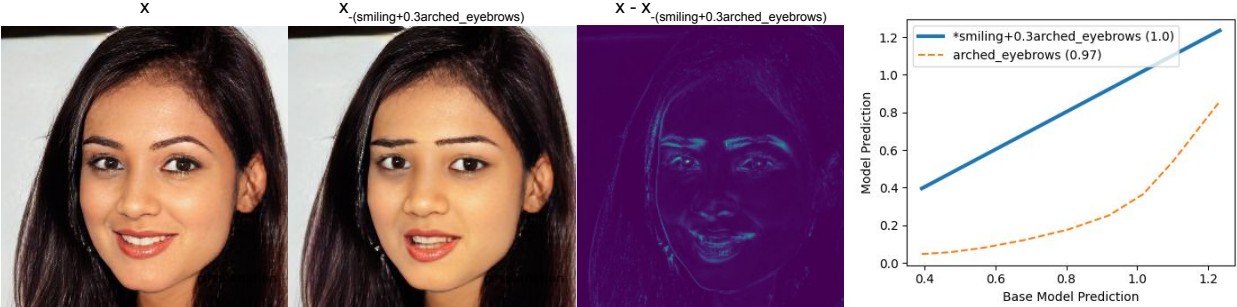

(b) CF for a modified *smiling* classifier which has been combined with an *arched_eyebrows* classifier.

Figure 4: Example of detecting a spurious correlation in a biased base classifier. The classifier is biased with arched eyebrows and this is observed in the alignment plot as well as in the counterfactual image. The relative change is now 0.97 compared to 0.01 for the unchanged *smiling* classifier.

can be induced in the classifier by composing classifiers:

$$f_{biased}(x) = f_{smiling}(x) + 0.3 f_{arched\_eyebrows}(x) \tag{4}$$

The CF alignment plot for the base *smiling* classifier predictions in Fig. 4a show that arched_eyebrows does not change and we can confirm this in the CF image. The resulting biased classifier can be observed using the *arched_eyebrows* features in Fig. 4b both visually as well as in the CF alignment plot. The CF alignment plot shows that the prediction of *arched_eyebrows* now changes and is aligned with *smiling*.

## 4.4    Analysis of robust optimization methods

An application of CF alignment is to assess the effectiveness of bias mitigation techniques. Various biased training settings have been developed to induce spurious correlations with sensitive attributes in models and then employ bias mitigation strategies to counter them. The experiments we present are designed to demonstrate that measuring the CF alignment with sensitive attributes can measure the impact of these methods.

To properly compare these methods, experiments are grouped based on the specific training configuration used so each baseline is unique to that setting. For this reason, these models cannot be compared side by side for the same dataset.

The bias mitigation methods we use are selected because their authors make their code available to generate pre-trained models that can be studied post-hoc. Using only pre-trained models further demonstrates the utility of our proposed approach where our method is versatile enough to evaluate these existing pre-trained models.

Experiments in this section are performed on the held out test datasets and selected to be balanced such that there is a balanced distribution of positive and negative examples with the sensitive attribute (e.g. in

the CelebA dataset, 1/4 of the samples are labeled to have *blond_hair* and be *male*, 1/4 *blond_hair* and be not *male*, 1/4 not *blond_hair* and be not *male*, and 1/4 not *blond_hair* and be *male*).

During classification experiments, the samples evaluated are selected to be inversely correlated with the sensitive attribute to cause the most discrepancy in performance (e.g. Samples are selected that are labeled *blond_hair* and *male* as well as not *blond_hair* and not *male*).

### 4.4.1 GroupDRO on Waterbirds

Work on GroupDRO Sagawa et al. (2020) constructed the Waterbirds task such that there is a spurious correlation between birds (from the CUB dataset (Wah et al., 2011)) with backgrounds that contain water or land.

Three models are evaluated for this experiment, a baseline model which was induced to rely on spurious correlations, a model trained with Group Distributionally Robust Optimization (DRO) (Sagawa et al., 2020) which aims to minimize error over groups (which are known during training, in this case the background class), a model using Just Train Twice (JTT) (Liu et al., 2021) which uses a two stage approach that boosts misclassified training examples in a second training cycle. To perform CF alignment, classifiers are trained to predict land and water using the samples from the waterbirds dataset labeled as those respective classes.

A limitation of this analysis is the limited latent variable model used (A VQ-GAN trained on Open Images (Krasin et al., 2017)) that has difficulty modulating features (likely because its training domain is broad). A model that is better at representing birds would be able to generate better counterfactuals that are easier to interpret.

In Figure 5 two models are evaluated on the same image illustrating a difference in CF alignment results to a classifier that predicts the background. The relative change with the background classifier is much higher for the baseline model than the model trained with DRO (0.35 vs 0.25) indicating that training with GroupDRO prevented the models reliance on features associated with the background classifier. A magnified view of the counterfactuals of the bird's head indicate nostril size being reduced and colors becoming brighter in the model trained with DRO. This may indicate larger nostril sizes are associated with water birds which aligns with the family of waterbirds, Procellariiformes, having an enlarged nasal gland at the base of the beak for secreting salt water (Ehrlich et al., 1988). While the mallard in the image is not a Procellariiforme, nostril size may be a feature used by the classifier in general as many in the waterbirds are seabirds.

Aggregate results over 4097 samples from the test set of the waterbirds dataset reveals a reduction in mean relative change using DRO shown in Table 1. This indicates that the features used by the baseline classifier overlap the land and water classifier and these spurious correlations were reduced when utilizing DRO. We observe that this improved model, with a lower relative change, achieves a higher classification accuracy.

| Target | Relative change ↓ w/water classifier | Relative change ↓ w/land classifier | Classification ↑ AUC |
|---|---|---|---|
| Waterbird (baseline) | 0.35±0.02 | -0.36±0.03 | 0.69 |
| Waterbird (groupdro) | **0.25±0.00** | **-0.22±0.03** | **0.92** |
| Waterbird (jtt) | 0.42±0.03 | -0.43±0.00 | 0.65 |

Table 1: Aggregate relative change metrics for models trained on the waterbirds test dataset (N=4097). A reduction in relative change with the land and water background classifiers indicates DRO has reduced the spurious correlation.

### 4.4.2 GroupDRO on CelebA

GroupDRO models are also evaluated on the CelebA dataset on the task of Blond_Hair and the sensitive attribute Male. JTT is not evaluated on this task because we were unable to generate pre-trained weights using the provided code due to memory issues that we were unable to resolve.

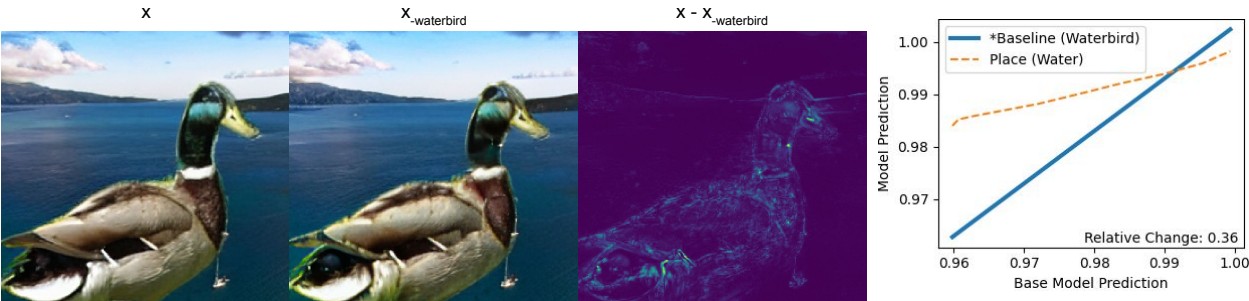

(a) Evaluating a baseline model trained that is expected to use spurious correlations.

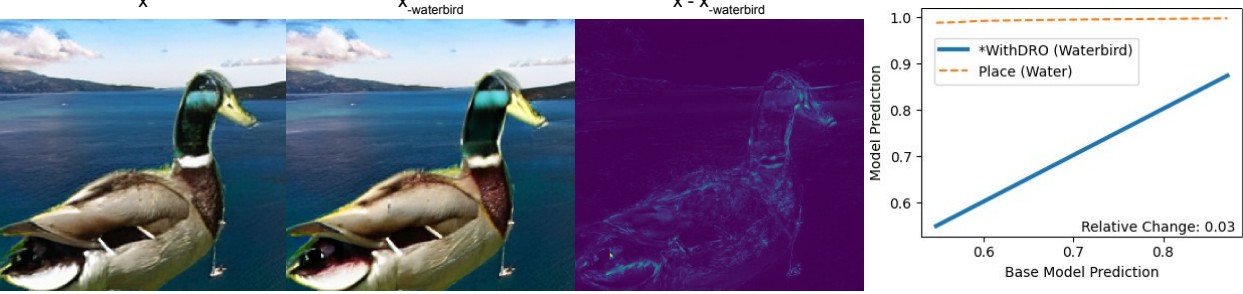

(b) Evaluating a model trained using Distributionally Robust Optimization (DRO) to reduce the spurious correlation with the background.

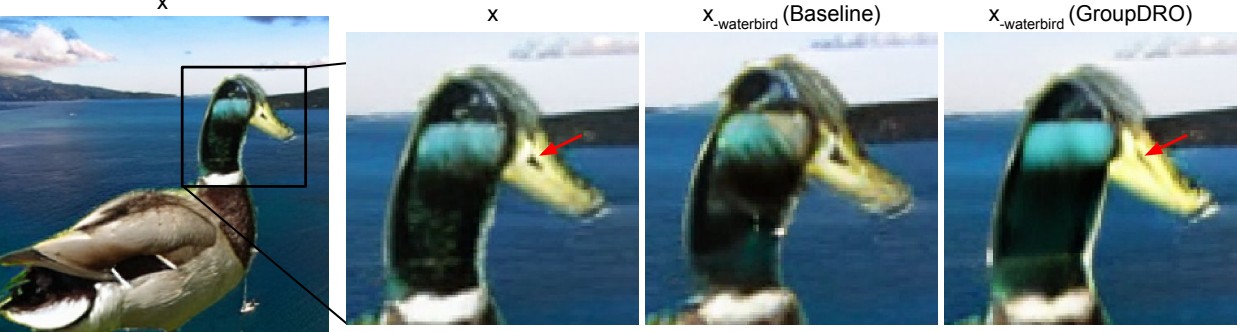

(c) A magnified view of the birds head which shows differences in nostril size and head color.

Figure 5: Counterfactuals generated for models trained on the waterbirds dataset together with CF alignment plots. (a) and (b) The relative change between the waterbird classifier and the background classifier is shown in the lower right of the plot. (c) The nostril size is reduced in the counterfactual for the DRO model indicating that a larger nostril size is associated with waterbirds.

In Table 2 we can see the relationship between Blond_Hair and Male is negative in the baseline and then is reduced by the GroupDRO training and becomes slightly positive. This reduction results in a higher classification accuracy on the CelebA test set.

| Target | Relative change ↓ w/male classifier | Classification ↑ AUC |
|---|---|---|
| Blond_Hair (baseline) | -0.052±0.01 | 0.91 |
| Blond_Hair (groupdro) | **0.025±0.00** | **0.95** |

Table 2: Methods evaluated on models that predict Blond_Hair trained on the CelebA dataset. Evaluations are performed on 1024 samples from the test set.

### 4.4.3 FLAC on CelebA

The work Fairness-Aware Representation Learning by Suppressing Attribute-Class Associations (FLAC) (Sarridis et al., 2023) uses pairwise similarity between the between the model representation and the representation of a bias-capturing classifier that predicts the sensitive attribute. The bias capturing classifier acts as a proxy for labels and provides embeddings that can be used to regularize the feature space of the classifier we are training. Their approach is to minimize the difference, in feature space, between samples with different sensitive attributes and the same target label as well as increase the difference between samples with the same sensitive attributes and the different target label. In Table 3 the relative change is shown to be reduced and the classification performance improves.

| Target | Alpha | Relative change ↓ w/male classifier | Classification ↑ AUC |
|--------|-------|-------------------------------------|----------------------|
| Blond_Hair | 0 | -0.081±0.016 | 0.77 |
| Blond_Hair | 30000 | **-0.039±0.009** | **0.93** |
| Blond_Hair | 60000 | -0.051±0.010 | 0.92 |

Table 3: Aggregate relative change of models trained with FLAC on 1024 samples from the CelebA test set. The strength of this regularization is controlled with an $\alpha$ parameter.

## 5 Limitations

There are limitations and challenges to CF generation (Verma et al., 2020) where bias can also exist in the CF generation method which can limit the features able to be modulated and prevent spurious features from being detected.

Spurious alignment: This situation can occur if the downstream classifier ($f_1$) also has the same spurious features as the base classifier ($f_b$). If this is the case, then both of their output would change if the spurious features were modulated causing these classifiers to appear aligned. For example, if a base classifier predicts *cow* but spuriously uses the presence of a beach to produce a low prediction. A CF generated for this classifier may change the background of a farm to a beach. If we check the alignment with a classifier that predicts *chicken* (but this chicken classifier also reduces its prediction if the image contains a beach) then these classifiers would appear aligned and the relative change would be positive. These classifiers are in fact aligned based on feature usage but just because they just use the same spurious features. This may be similar to a false positive but it isn't false, the classifiers are aligned.

For this reason, in this work we reference the classifier target names (e.g. *mouth_slightly_open*) instead of the concept that it is assumed to represent. One solution to avoid these edge cases is to utilize multiple classifiers that are trained to predict the same concept using multiple training datasets so on average they predict the correct concept.

False negative alignment: This situation can occur if the downstream classifier ($f_1$) predicts using features that are "opposite" of the base classifier ($f_b$). For example "opposite" can mean the base classifier ($f_b$) predicts *cow* using only the presence of absence of a beach while the downstream classifier ($f_1$) predicts *beach* using only the presence of absence of a cow. The CF for the base classifier changes the background from a beach to something else but this doesn't change the prediction of the downstream classifier ($f_1$) so there is no alignment, the relative change would be 0. This case could be considered a false negative because we didn't find alignment with a *beach* classifier but the *beach* classifier was flawed. To address an issue like this we could simply inspect the CFs that are being generated with domain experts as well as vary the downstream classifier's training data.

Furthermore, although we are optimistic that our method will generalize to additional domains, we focus our current analysis on face-attribute classification and waterbirds. Although we do not demonstrate the generalizability of our method to additional domains, using face-attribute classification is a deliberate choice enabling qualitative evaluation, in addition to our quantitative evaluation. Additionally, since CF generation

relies on a separate autoencoder, improving the representational capability of the autoencoder may improve the fidelity of the CF generation.

## 6 Conclusion

In this work we propose counterfactual (CF) alignment along with the relative change metric §3. We demonstrate that this method enables us to reason about the feature relationships between classifiers in aggregate and to locate specific examples where the spurious correlations are used. These claims are supported with an analysis of face-attribute classifiers that identify expected and unexpected spurious correlations.

We observe that if correlated attributes are presented to the classifier, this does not cause feature usage to be correlated §4.1. We also observe that if uncorrelated attributes are presented to the classifier, the classifier may still construct spurious feature relationships.

The validity of the CF alignment method is confirmed by inducing and quantifying spurious correlations via additive composition §4.3. Classifiers are composed together to create a classifier with a known spurious correlation and then this is observed using CF alignment.

We then explore classifiers trained using robust optimization methods to demonstrate the applicability to black box classifiers and to provide more visibility into what these methods achieve §4.4.

Overall, the proposed approach may serve as an end-to-end or human-in-the-loop system to automatically detect, quantify, and correct spurious correlations for image classification tasks that lead to biased classifier outputs.

## 7 Acknowledgments

We would like to thank Stanford University and the Stanford Research Computing Center for providing computational resources and support that contributed to these research results.

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

# A    Appendix

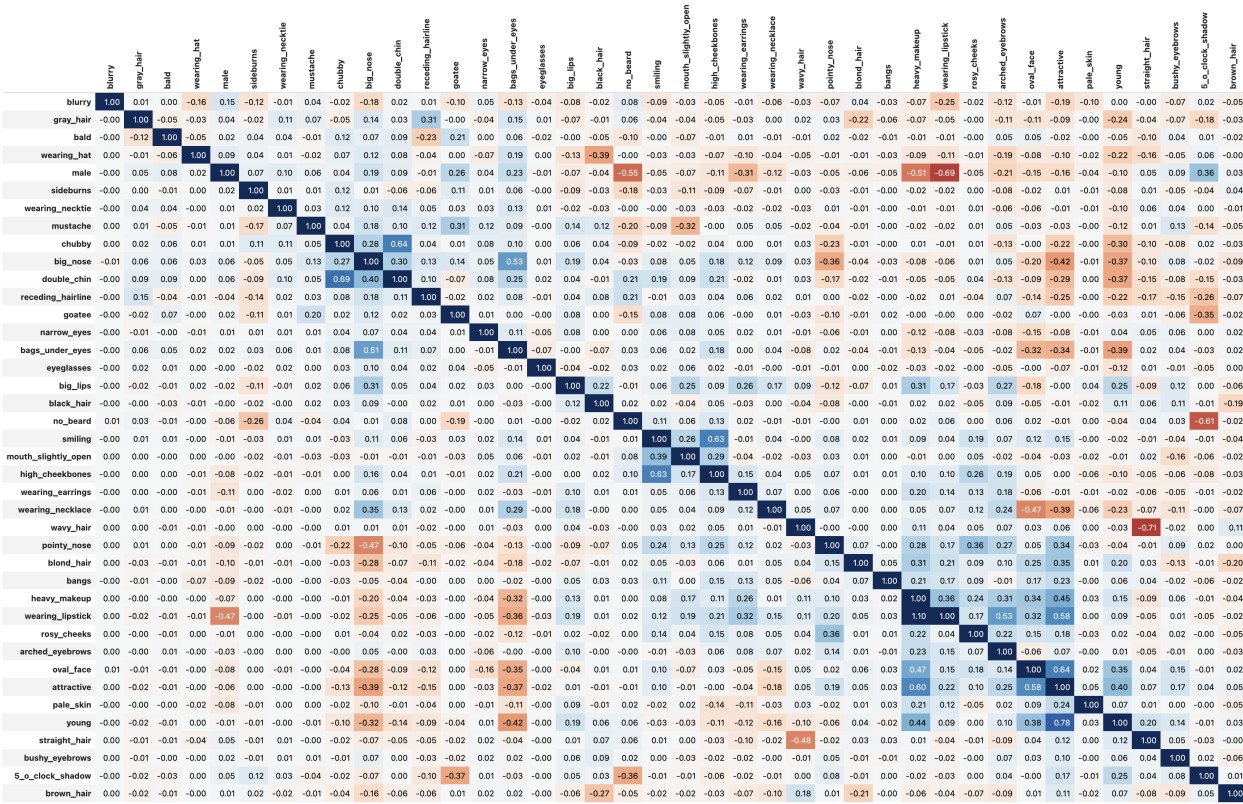

Figure 6: The complete matrix of relative changes for all classifiers. Ordering is determined by clustering to group similar classifiers.

## A.1 Comparison with saliency maps

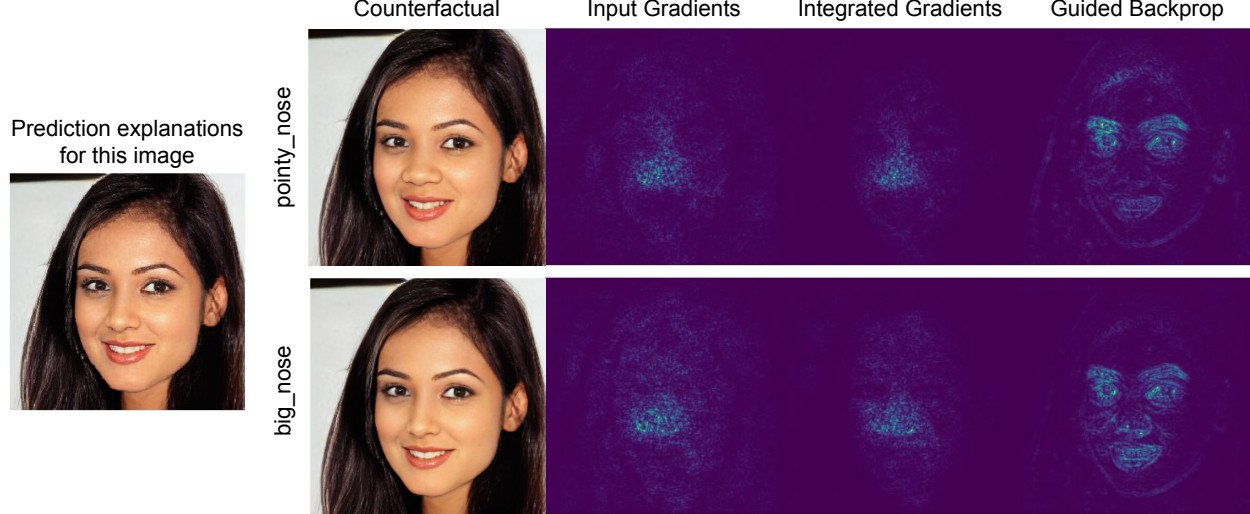

Figure 7: Example of saliency maps failing to provide meaningful localization when the concepts overlap with each other. Here, saliency map methods localize around the nose which doesn't provide the ability to distinguish between a big or pointy nose prediction.

Saliency maps for feature attribution could potentially be used to perform a similar analysis by looking at relationships between their generated heatmaps. Here, we present an example that demonstrates their limitation. The attribution methods input gradients, integrated gradients (Sundararajan et al., 2017), and guided backprop (Springenberg et al., 2015) are used to explain the classifier predictions of *pointy_nose* and *big_nose*.

In Fig. 7 the salient areas for *pointy_nose* and *big_nose* are the same and cannot be disambiguated. The saliency maps only present information as pixel importance which overlaps because both classifiers use features on the nose. Due to these very similar heatmaps, comparing them would not allow us to conclude that the classifiers are using different features. We observe in Fig. 2 that these two classifiers have an inverse relationship. Using CF alignment, we can gain a deeper understanding of what features the model is using, and we can better reason about why the decision was made.

## A.2    Comparison with other CF generation methods

Using the biased classifier from §4.3, we evaluate whether another counterfactual (CF) generation methods can capture its bias. As a baseline, we use the Wachter et al. (2017) method, which generates a new $x'$ that minimizes the classifier's prediction as well as the proximity to the input $x$. The objective is as follows:

$$\underset{x'}{\operatorname{argmin}} \ \lambda \cdot d(x, x') + (f(x') - y')^2 \tag{5}$$

The distance function $d$ is chosen to be the Frobenius norm in order to match pixels in the image. While this approach was not designed for images it is still able to change the base classifiers prediction in our experiments. The CF generated with this method is shown in Figure 8 and does not appear to generate features that capture alignment with the classifier's bias. This method operates in a similar way to adversarial examples which introduce imperceptible changes.

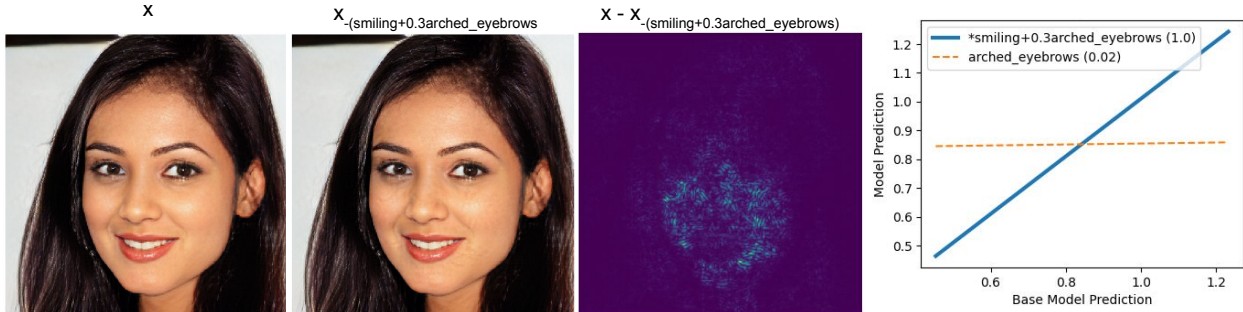

Figure 8: Comparison with the Wachter counterfactual generation method. A *smiling* classifier with a known bias of *arched_eyebrows* is used to observe if this bias can be detected. The Wachter counterfactual doesn't appear to modulate the input image in a human interpretable way, instead many small swirls appear on the face which change the prediction for *smiling* but not for *arched_eyebrows*.

### A.3 Rectifying bias by composing classifiers

This section demonstrates the use of CF alignment to fix model bias in classifiers. We can reuse the relative change between model predictions as a loss function that can be minimized.

This section just serves as an example to better understand a use of the CF alignment idea. We don't claim this method is competitive to other methods which correct model bias and spurious correlations. Related approaches include averaging model weights Wortsman et al. (2022) or methods of invariant optimization Sagawa et al. (2020); Krueger et al. (2021); Rieger et al. (2020); Zeng et al. (2023) and approaches for balanced dataset sampling Singh et al. (2021). Composing classifiers can change their response to specific samples that contain relevant features as shown in §4.3. By using this additive composition approach we avoid the variance of model training which makes it easy and reproducible to perform experiments.

First, we demonstrate a single example of correcting bias. In Fig. 9, the CF image and CF alignment plot for the *heavy_makeup* base classifier show unexpected reductions in lip size. Composing the model with the *big_lips* classifier using a negative coefficient causes the classifier to lower any feature change focused on the lips as this would increase the prediction of the composed classifier (where the goal of the CF is to decrease it). We can also visually observe other features that change, such as skin color, bushy eyebrows, and the amount that the mouth is open.

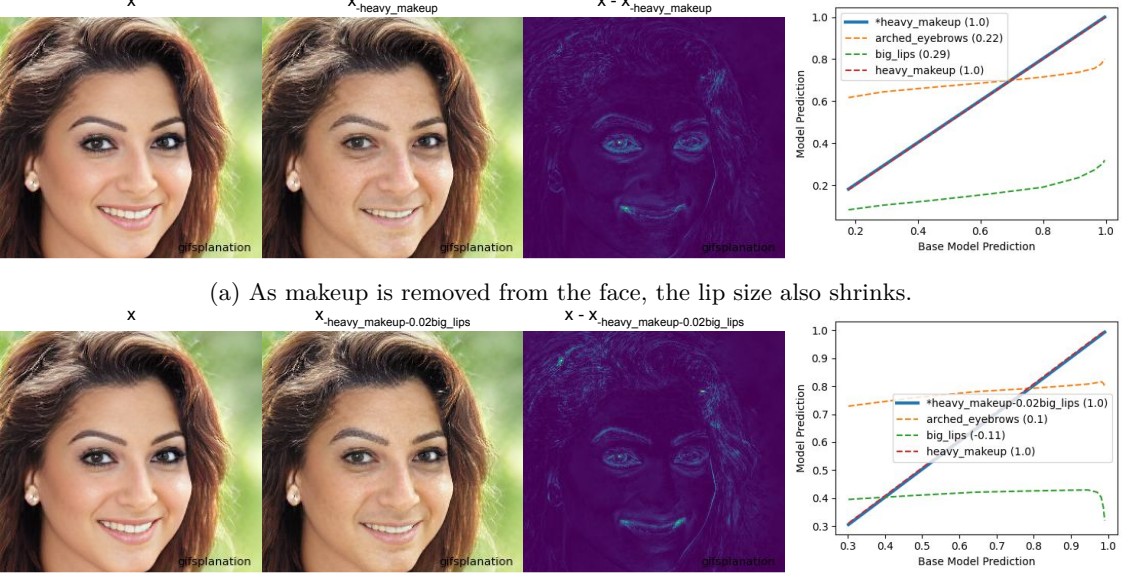

(a) As makeup is removed from the face, the lip size also shrinks.

(b) The lip size remains the same as the makeup is removed.

Figure 9: An example of a *big_lips* spurious correlation being corrected for the *heavy_makeup* classifier.

Next, this unbiasing is scaled up to a larger number of samples and classifiers. A collection of 12 classifiers that contain varying spurious relationships to a *big_nose* base classifier (shown in Fig. 10a) are modified to remove their bias until their relative change is as close to 0 as possible. This experiment is performed with a train, validation, and hold out test set in order to demonstrate the generalization of this unbiasing process to unseen data.

The classifiers are modified using a single coefficient ($\beta$) such that the resulting classifier is

$$f'_{target}(x) = f_{target}(x) + \beta f_{big\_nose}(x) \tag{6}$$

Optimization to compute each $\beta$ is performed using a pseudo gradient descent where the gradient is approximated by the mean relative change between the target classifier and the *big_nose* classifier. By subtracting the relative change from $\beta$, scaled by a learning rate, the relative change ($\psi$) with that classifier will be

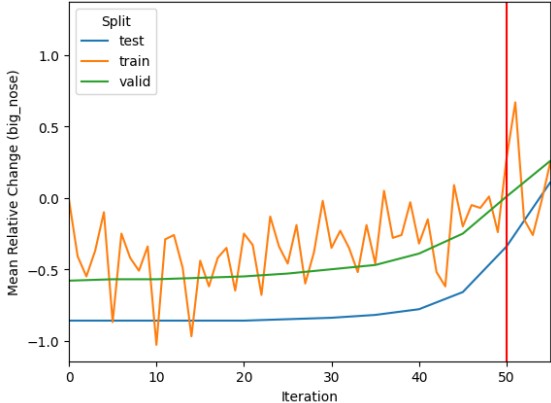

Figure 10: Training curves showing mean relative change during the optimization of a classifier with respect to the *big_nose* classifier during each iteration. The red vertical line is the early stopping point based on the validation set. This demonstrates how bias can be quantified and corrected using CF alignment.

reduced. All together our training objective (including momentum) is

$$\beta_n = 0.001\psi(f_{target}, f_{big\_nose}) + 0.1\beta_{n-1} \tag{7}$$

We find that using small minibatches of 10 samples works well because the computation time for each CF can take over 1 second on a GPU. Additionally, training with samples which induce a small change in the base classifier during the CF generation process can be challenging. As the classifiers are modified, this base change is reduced, which causes the relative change to become more erratic and prevents the optimization from converging. To prevent this, we use samples with a base change $> 0.6$.

The resulting training metrics in Fig. 10 show model biases being minimized similar to a differentiable training loss. Looking at the mean relative change allows us to summarize the bias of the model with respect to the *big_nose* classifier who's relationship we aim to remove. We observe a bias generalization gap between the train, valid, and test sets, indicating some degree of overfitting. Early stopping (red vertical line) is used on the validation set to determine the optimal parameters. We use momentum during training to average over the noise.

The resulting reduction in bias is shown per classifier before and after training in Fig 11. 10 of the 12 classifiers being optimized have the bias considerably reduced (closer to 0) except for *attractive* and *big_nose*, which could be the result of overlapping features used by both classifiers that cannot be changed.

**A) Baseline**

| | mouth_slightly_open | big_nose | male | bushy_eyebrows | arched_eyebrows | smiling | high_cheekbones | pointy_nose | wearing_lipstick | heavy_makeup | oval_face | attractive | young |
|---|---|---|---|---|---|---|---|---|---|---|---|---|---|
| mouth_slightly_open | 1.00 | -0.20 | -0.00 | -0.04 | -0.16 | 0.52 | 0.21 | 0.04 | 0.03 | 0.03 | 0.24 | 0.04 | -0.07 |
| big_nose | 0.08 | 1.00 | 0.00 | 0.82 | -0.01 | 0.61 | 0.55 | -0.42 | -0.06 | -0.34 | -0.36 | -0.56 | -0.02 |
| male | -0.01 | 0.24 | 1.00 | 0.11 | -0.02 | -0.02 | -0.09 | -0.09 | -0.39 | -0.20 | -0.03 | -0.20 | -0.01 |
| bushy_eyebrows | 0.07 | -0.05 | 0.00 | 1.00 | 0.18 | -0.01 | -0.06 | 0.06 | 0.00 | 0.00 | -0.03 | 0.12 | 0.19 |
| arched_eyebrows | -0.01 | 0.25 | 0.00 | 0.02 | 1.00 | -0.00 | 0.00 | 0.24 | 0.00 | 0.02 | -0.33 | -0.06 | -0.01 |
| smiling | 0.33 | 0.25 | -0.00 | -0.00 | 0.05 | 1.00 | 0.90 | 0.02 | 0.00 | 0.01 | 0.12 | 0.13 | -0.27 |
| high_cheekbones | 0.02 | 0.05 | -0.01 | -0.10 | 0.38 | 0.04 | 1.00 | 0.20 | 0.03 | 0.11 | -0.06 | -0.01 | -0.16 |
| pointy_nose | -0.05 | -0.86 | -0.00 | 0.08 | -0.01 | 0.29 | 0.28 | 1.00 | 0.00 | 0.03 | 0.10 | 0.49 | -0.00 |
| wearing_lipstick | -0.05 | -0.06 | -0.41 | -0.02 | 0.41 | -0.01 | 0.02 | 0.14 | 1.00 | 1.26 | 0.06 | 0.52 | -0.03 |
| heavy_makeup | 0.27 | -0.03 | -0.01 | 0.05 | 0.14 | 0.03 | 0.24 | 0.11 | 0.39 | 1.00 | 0.31 | 0.12 | 0.00 |
| oval_face | 0.11 | -0.41 | -0.00 | 0.16 | -0.20 | 0.36 | 0.22 | 0.24 | 0.00 | 0.01 | 1.00 | 0.56 | 0.01 |
| attractive | -0.04 | -0.49 | -0.08 | 0.20 | 0.08 | 0.09 | -0.09 | 0.22 | 0.29 | 1.04 | 0.67 | 1.00 | 0.24 |
| young | -0.02 | -0.18 | 0.38 | 0.01 | -0.02 | -0.02 | -0.19 | -0.03 | -0.25 | -0.05 | 0.05 | 0.56 | 1.00 |

**B) Optimized**

| | mouth_slightly_open | big_nose | male | bushy_eyebrows | arched_eyebrows | smiling | high_cheekbones | pointy_nose | wearing_lipstick | heavy_makeup | oval_face | attractive | young |
|---|---|---|---|---|---|---|---|---|---|---|---|---|---|
| mouth_slightly_open | 1.00 | -0.01 | 0.00 | -0.02 | -0.13 | 0.56 | 0.23 | 0.02 | 0.01 | -0.01 | -0.02 | -0.03 | -0.09 |
| big_nose | 0.08 | 1.00 | 0.00 | 0.82 | -0.01 | 0.61 | 0.55 | -0.42 | -0.06 | -0.34 | -0.36 | -0.56 | -0.02 |
| male | -0.01 | -0.00 | 1.00 | 0.09 | -0.04 | -0.02 | -0.10 | -0.05 | -0.34 | -0.16 | 0.00 | -0.12 | -0.00 |
| bushy_eyebrows | 0.07 | -0.05 | 0.00 | 1.00 | 0.18 | -0.01 | -0.06 | 0.06 | 0.00 | 0.00 | -0.03 | 0.12 | 0.19 |
| arched_eyebrows | -0.01 | 0.00 | -0.00 | 0.01 | 1.00 | -0.01 | 0.00 | 0.31 | 0.01 | 0.06 | -0.16 | 0.08 | -0.00 |
| smiling | 0.44 | -0.01 | -0.00 | 0.00 | -0.08 | 1.00 | 0.79 | 0.02 | 0.01 | 0.02 | 0.19 | 0.35 | -0.13 |
| high_cheekbones | 0.02 | 0.01 | -0.02 | -0.12 | 0.40 | 0.04 | 1.00 | 0.20 | 0.04 | 0.14 | -0.03 | -0.01 | -0.16 |
| pointy_nose | -0.07 | 0.11 | 0.00 | 0.09 | 0.01 | 0.38 | 0.69 | 0.94 | -0.00 | -0.01 | -0.07 | -0.13 | -0.12 |
| wearing_lipstick | -0.05 | -0.00 | -0.40 | 0.00 | 0.43 | -0.01 | 0.03 | 0.11 | 1.00 | 1.26 | 0.06 | 0.48 | -0.03 |
| heavy_makeup | 0.27 | -0.03 | -0.01 | 0.05 | 0.14 | 0.03 | 0.24 | 0.11 | 0.39 | 1.00 | 0.31 | 0.12 | 0.00 |
| oval_face | -0.06 | 0.04 | -0.00 | 0.17 | -0.14 | 0.41 | 0.43 | -0.22 | 0.00 | 0.01 | 0.99 | 0.45 | 0.01 |
| attractive | -0.04 | -0.49 | -0.08 | 0.20 | 0.08 | 0.09 | -0.09 | 0.22 | 0.29 | 1.04 | 0.67 | 1.00 | 0.24 |
| young | -0.02 | -0.02 | 0.75 | 0.01 | -0.01 | -0.02 | -0.20 | -0.07 | -0.49 | -0.13 | 0.04 | 0.54 | 1.00 |

Figure 11: CF alignment matrices before and after optimization to remove the bias for *big_nose* on a hold out test set. The figure shows a dramatic reduction in relative change for most classifiers and *big_nose* with limited residual impact on other relationships.

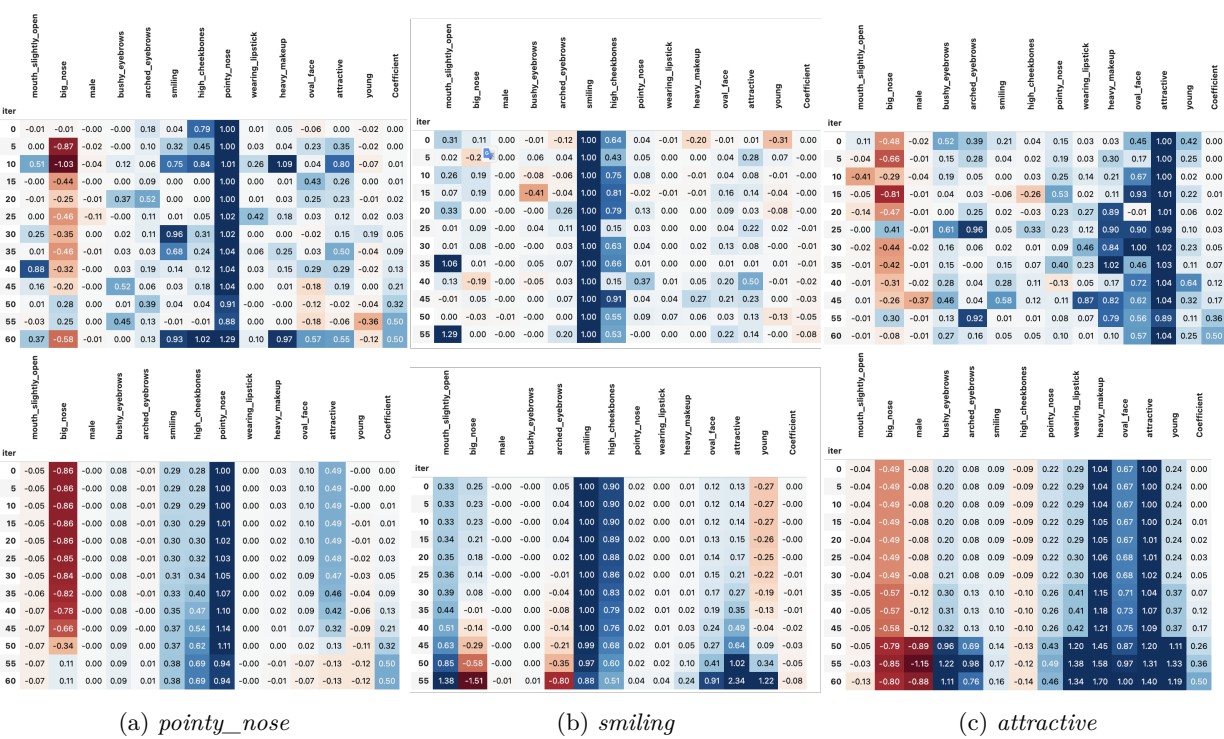

(a) *pointy_nose*        (b) *smiling*        (c) *attractive*

Figure 12: Visualization of the training from §A.3 reducing the bias of *big_nose*. The model being evaluated is $f'_{target}(x) = f_{target}(x) + \beta f_{big\_nose}(x)$ where $\beta$ is specified by the coefficient column. The metric here is relative change computed between the $f'_{target}(x)$, where *target* is specified in the caption, and the classifier specified by the column header. The top row is from the training set performed on batches (which explains the variance) and the second row is the results on the entire validation set. The row of each table is for a specific iteration. The iterations proceed downward. The training of *attractive* shown here does not result in a debiased model, the relative change diverges from 0 indicating a failed training.

