# OpenReview forum: "Identifying Spurious Correlations using Counterfactual Alignment"
_TMLR — Accepted by TMLR_

### Review · Reviewer_zg5u · 2024-11-01

**Summary Of Contributions:**

The authors propose a method called *counterfactual alignment* for detecting spurious features in image classification via a novel *relative change metric* . This method considers a pair of classifiers: a base classifier and a downstream classifier. In broad strokes, a counterfactual explanation in the latent representation of an auto-encoder is run on the base classifier and compared to the change in the prediction of the downstream classifier on the output counterfactual image. A spurious feature is detected if this relative change in the prediction of the classifiers deviates far from $1$. The method is assessed empirically on CelebA and Waterbird data sets, where the method is demonstrated to detect spurious features reliably and in a localized way. This lies in contrast to other explainable AI methods such as saliency maps, which fail to localize spurious features. In addition, the proposed method indicates that robust methods like GroupDRO, JTT, and FLAC are indeed less reliant on spurious features.

**Audience:**

Yes

**Broader Impact Concerns:**

No broader impact concerns

**Claims And Evidence:**

No

**Requested Changes:**

Critical requested changes are highlighted by the comment *(critical)*.

**Abstract**:

1) *face-attribute face-attribute*. This seems to be a typo.

**1. Introduction**:

1) *Shortcut learning can
lead to models making decisions based on factors not aligned with expectations of the model creator.* What is a *model creator*? Please clarify.

2) *of an image (Pearl, 2009)* I believe that this is not the right reference. Pearl's construction of counterfactuals is interventional. It seems the authors instead use *counterfactual explanations* [1] , which can be derived from (non-interventional) *backtracking counterfactuals* (see [2]). Please clarify or change reference.

3) *using a singular fixed autoencoder as a reference point.* What is a *singular autoencoder*? Please clarify.

4) *This shared feature usage can prompt us to investigate unexpected relationships*. So if a feature is used by multiple models, that means there are unexpected relationships? I cannot follow. What is even meant by *unexpected relationships*? And why is that the case? Please clarify, maybe by providing a concrete example.

5) *We propose CF alignment to reason about the feature relationships between classifiers* It is not clear to me what is meant by *feature relationships between classifiers*. Please clarify.

**3.1 Relative Change**:

1) (critical) *This decision was motivated by the observation that correlation metrics occasionally yielded false positives when only a slight change in the prediction of $f_1$ occurred compared to the base classifier $f_b$*. I do not understand this explanation. It seems to me that correlation measures differences between classifiers on a population level, in contrast to this relative change metric, which is instance-specific. However, a population-level variant of the proposed metric is considered in section 4.1. It would be great if this could be discussed in greater detail.

**2. Experiments**:

1) (critical) *The pre-trained face classifiers used in this work are sourced from...* How do these classifiers differ from one another? By random seed or by architecture? I believe that this is quite crucial.

2) (critical) What I am missing in all experiments is a clear description of which model is used as base classifier and which one(s) is/are used as downstream classifier(s).

**Figure 2**:

I am not sure if I understand this figure correctly. Does that mean that the target label of the base classifier differs from the target label of the downstream classifier? How would this help in finding spurious features?

**4.1 Aggregate statistics over a dataset**

1) *N=400 per class*. What is $N$? The number of different classifiers?

**4.2 Studying specific examples**

1) (critical) *inverse alignment with big_nose* What is meant by *inverse alignment*? How is *alignment* even defined? If the authors introduce a term, it would be great to state explicitly what it means. It would be great if the authors could write something like *We say that classifier 1 and classifier 2 are aligned if ...* and analogously for *inverse aligned*. From Figure 3, I would guess that *alignment* means that the relative change metric is monotonically inreasing for the two classifiers (?).

2) *rosy_cheeks is a common aligned classifier which does not appear to have an obvious reason* I do not understand this sentence. What is meant by *obvious reason*? Also, as already mentioned, *aligned classifier* is not defined.

**4.5.2 GroupDRO on CelebA**

1) *we were unable to generate pre-trained weights using the provided code due to memory issues that we were unable to resolve.* Not a requested change, but I just would like to appreciate the honesty of the authors here. Quite refreshing.

**6 Conclusion**

1) *We also observe that if uncorrelated attributes are presented to the classifier, the classifier may still construct spurious feature relationships.* This is an intersting observation. Do the authors have a hypothesis for why this happens?

[1] Wachter, Sandra, Brent Mittelstadt, and Chris Russell. "Counterfactual explanations without opening the black box: Automated decisions and the GDPR." Harv. JL & Tech. 31 (2017): 841.

[2] Kladny, Klaus-Rudolf, et al. "Deep Backtracking Counterfactuals for Causally Compliant Explanations." Transactions on Machine Learning Research (2024).

**Strengths And Weaknesses:**

**Strengths:**

1) The study considers a relevant and long-standing problem (spurious features).

2) Under a certain assumption (see weakness 1), the proposed approach seems reasonable. The experimental results are convincing.

3) Plenty of experiments on proper computer vision data sets, including future release of the source code for reproducibility.

4) The manuscript is written in simple language and notation, making it accessible to a broad audience with diverse backgrounds in machine learning.

**Weaknesses:**

1) It seems the work hinges on the strong assumption that this *base classifier* is not affected by spurious features. This assumption should be made explicit and discussed critically. For instance, if both the *base classifier* and the *downstream classifier* use the presence of a beach to not predict the *cow* class, the relative change metric will not detect this spurious feature.

2) The writing must be improved. Major parts of the paper are not clear to me (see weakness 3 and requested changes) and it may well be that my understanding of the proposed method is incorrect. Specifically, it would be great if the authors could provide more concrete examples and clear definitions.

3) It is still not clear to me what the advantage of this *relative change metric* is in comparison to other metrics such as correlation of the change between base and downstream classifier. The *relative change metric* is stated as the main contribution of the study, so there should be a greater discussion around this aspect.

---

> ### Author Response · Authors · 2024-12-12
> **Response**
>
> We thank the reviewer for their feedback!
>
> > 1. Introduction:
> Shortcut learning can lead to models making decisions based on factors not aligned with expectations of the model creator. What is a model creator? Please clarify.
>
> This was rephrased to "human that created the model"
>
> > of an image (Pearl, 2009) I believe that this is not the right reference. Pearl's construction of counterfactuals is interventional. It seems the authors instead use counterfactual explanations [1] , which can be derived from (non-interventional) backtracking counterfactuals (see [2]). Please clarify or change reference.
>
> Thanks for pointing this out, we have changed the reference to Wachter 2017.
>
> > using a singular fixed autoencoder as a reference point. What is a singular autoencoder? Please clarify.
>
> This is rephrased to "a single autoencoder as a reference point". This was just meant to say that there is only one AE used while we vary the classifiers.
>
> > This shared feature usage can prompt us to investigate unexpected relationships. So if a feature is used by multiple models, that means there are unexpected relationships? I cannot follow. What is even meant by unexpected relationships? And why is that the case? Please clarify, maybe by providing a concrete example.
>
> The text was rephrased to:
> "This shared feature usage can prompt us to investigate if this relationship is unexpected. Such as if a classifier that predicts big noses should share features with a classifier that predicts arched eyebrows."
>
> > - We propose CF alignment to reason about the feature relationships between classifiers It is not clear to me what is meant by feature relationships between classifiers. Please clarify.
>
> We mean the features used to make a prediction. This phrasing was changed to "feature usage relationships" for clarity.
>
>
> > 3.1 Relative Change:
> (critical) This decision was motivated by the observation that correlation metrics occasionally yielded false positives when only a slight change in the prediction of occurred compared to the base classifier. I do not understand this explanation. It seems to me that correlation measures differences between classifiers on a population level, in contrast to this relative change metric, which is instance-specific. However, a population-level variant of the proposed metric is considered in section 4.1. It would be great if this could be discussed in greater detail.
>
> In section 4.1 we consider the average relative change over all the samples in a dataset. The wording was updated to make this more clear.
>
> > (critical) The pre-trained face classifiers used in this work are sourced from... How do these classifiers differ from one another? By random seed or by architecture? I believe that this is quite crucial.
>
> The face classifiers have the same architecture. The only difference between them is the last classifier layer weights. The waterbirds and land/water classifiers also have the same architecture but were trained independently on different data so the weights are different throughout the model. This is added to the text.

---

> ### Author Response · Authors · 2024-12-12
> **Response**
>
> > (critical) What I am missing in all experiments is a clear description of which model is used as base classifier and which one(s) is/are used as downstream classifier(s).
>
> The wording was changed to specify what the base classifier is in the figures and explanations.
>
> > Figure 2: I am not sure if I understand this figure correctly. Does that mean that the target label of the base classifier differs from the target label of the downstream classifier? How would this help in finding spurious features?
>
> We use the phrase classifier to represent a single prediction and not a multi target prediction. This was added to the paper:  Here we consider a classifier as predicting a single scalar value $f(x) \in [0,1]$. A multiclass classifier can also work if we consider one class vs all other classes to create a single output value.
>
> >How would this help in finding spurious features?
>
> A high relative change means when the CF is constructed from the base classifier it also changes the predictions of the downstream classifier, implying that there is shared feature usage.
>
> > 4.1 Aggregate statistics over a dataset
> N=400 per class. What is N? The number of different classifiers?
>
> N here is the number of image samples. This has been edited in the paper.
>
>
> > 4.2 Studying specific examples
> (critical) inverse alignment with big_nose What is meant by inverse alignment? How is alignment even defined? If the authors introduce a term, it would be great to state explicitly what it means. It would be great if the authors could write something like We say that classifier 1 and classifier 2 are aligned if ... and analogously for inverse aligned. From Figure 3, I would guess that alignment means that the relative change metric is monotonically inreasing for the two classifiers (?).
>
> Thank you for pointing this out. This text has been added to the section "CF Alignment Methodology": We say that two classifiers are aligned if both their predictions change negatively with a counterfactual generated for one of them, implying the features removed were used for both predictions. We say they are inverse aligned when one classifier's prediction is reduced and the other's prediction increased.
>
> > rosy_cheeks is a common aligned classifier which does not appear to have an obvious reason I do not understand this sentence. What is meant by obvious reason? Also, as already mentioned, aligned classifier is not defined.
>
> Here we mean obvious as in something that would be expected, as the authors are knowledgeable in the domain of human faces. The text was rewritten to "does not appear to have an intuitive reason (to the authors)"

---

> > ### Comment · Reviewer_zg5u · 2024-12-12
> >
> > I thank the authors for implementing most of my feedback. However, I have one final concern that I request the authors to consider in their revision: **Weakness 1**. I believe that this is quite a severe limitation, and I frankly cannot imagine practical scenarios where one would ever have such a non-spurious base classifier to begin with. For me to recommend accept, I request a *salient* paragraph that discusses weakness 1. Without such a paragraph, the claims of the manuscript are not supported in my view and hence the manuscript does not fulfill the criteria for acceptance.

---

> > > ### Author Response · Authors · 2024-12-13
> > > **Response**
> > >
> > > >1. It seems the work hinges on the strong assumption that this base classifier is not affected by spurious features. This assumption should be made explicit and discussed critically. For instance, if both the base classifier and the downstream classifier use the presence of a beach to not predict the cow class, the relative change metric will not detect this spurious feature.
> > >
> > > Yes, thanks for pointing this out! I believe you mean to focus on the downstream classifier. In this work we are trying to identify the spurious features used by the base classifier so it is expected that the base classifier uses spurious features.
> > >
> > > In the case of the beach used as a feature in a base classifier that predicts "no_cow", a CF that would reduce this prediction (to increase the prediction of cow) may change the background to a farm. If the downstream classifier was a "no_chicken" classifier that also would produce a high score given the presence of a beach may also have the score reduced as the beach is changed to a farm. So these classifiers would be considered aligned and have a positive relative change because they do share feature usage. However we could check the alignment with other classifiers that predict "beach" and "farm" to uncover what is going on.
> > >
> > > To your point, if the beach classifier used the presence of a cow to not predict beach and the cow exists in the input and CF image then the prediction wouldn't change and the relative change would be 0. We could debug this by looking at the input and CF image directly to see what is going on. We could also evaluate the performance of the downstream classifier to observe that the accuracy is bad.
> > >
> > >
> > > The following text was added to the paper to discuss this issue:
> > >
> > > "Spurious alignment: This situation can occur if the downstream classifier ($f_1$) also has the same spurious features as the base classifier ($f_b$). If this is the case, then both of their output would change if the spurious features were modulated causing these classifiers to appear aligned. For example, if a base classifier predicts \emph{cow} but spuriously uses the presence of a beach to produce a low prediction. A CF generated for this classifier may change the background of a farm to a beach. If we check the alignment with a classifier that predicts \emph{chicken} (but this chicken classifier also reduces its prediction if the image contains a beach) then these classifiers would appear aligned and the relative change would be positive. These classifiers are in fact aligned based on feature usage but just because they just use the same spurious features. This may be similar to a false positive but it isn't false, the classifiers are aligned.
> > >
> > > For this reason, in this work we reference the classifier target names (e.g. \emph{mouth\_slightly\_open}) instead of the concept that it is assumed to represent. One solution to avoid these edge cases is to utilize multiple classifiers that are trained to predict the same concept using multiple training datasets so on average they predict the correct concept.
> > >
> > > False negative alignment: This situation can occur if the downstream classifier ($f_1$) predicts using features that are "opposite" of the base classifier ($f_b$). For example "opposite" can mean the base classifier ($f_b$) predicts \emph{cow} using only the presence of absence of a beach while the downstream classifier ($f_1$) predicts \emph{beach} using only the presence of absence of a cow. The CF for the base classifier changes the background from a beach to something else but this doesn't change the prediction of the downstream classifier ($f_1$) so there is no alignment, the relative change would be 0. This case could be considered a false negative because we didn't find alignment with a \emph{beach} classifier but the \emph{beach} classifier was flawed. To address an issue like this we could simply inspect the CFs that are being generated with domain experts as well as vary the downstream classifier's training data."
> > >
> > > Please let us know if there is another case that we should discuss!

---

> ### Comment · Reviewer_zg5u · 2024-12-13
>
> I thank the authors for including this paragraph. I am not sure whether I understand the sentence *We could also evaluate the performance of the downstream classifier to observe that the accuracy is bad* correctly. I would just like to point out that I believe we need to be careful about such statements: In an in-distribution setting, classifiers that rely on spurious features will typically outperform ones that do not.
>
> All in all, I believe that the acceptance criteria are now fulfilled and I thank the authors for their diligent efforts. I will recommend accept. I wish the authors all the best for this and all future work!

---

> > ### Author Response · Authors · 2024-12-14
> > **Response**
> >
> > > we need to be careful about such statements: In an in-distribution setting, classifiers that rely on spurious features will typically outperform ones that do not.
> >
> > Yes I agree. I should have specified the use of external datasets or controlled datasets that aim to cause the model to fail in order to understand how it is working.
> >
> > > I will recommend accept. I wish the authors all the best for this and all future work!
> >
> > Thank you!

---

### Review · Reviewer_vier · 2024-11-11

**Summary Of Contributions:**

The paper introduces a novel method of Counterfactual generation based on perturbing the encoding of images using the gradient of a base classifier. This methodology is then used coupled with a new Relative Change metric to find specific instances of classifiers using spurious correlations to output their predictions. It also provides an analysis of specific examples of these spurious correlations. Finally, this methodology is also used with Robust Optimization methods to demonstrate their better generalization capabilities as spurious correlations are reduced.

**Audience:**

Yes

**Claims And Evidence:**

Yes

**Requested Changes:**

See weaknesses section.

**Strengths And Weaknesses:**

Strengths:
- CF Alignment methodology is elegant, rigorous, and can be reused for a number of tasks - also shows promising results
- Likewise, Relative Change metric is a flexible a reusable metric when comparing classifiers
- Extensive study of specific examples and comparison with other methods (e.g. Saliency maps) gives interesting insights on the behavior of classifiers when encountering CFs
- Link with Robust Optimization is interesting and unifies the whole framework

Weaknesses:
- I would have appreciated a further study and insights on the aggregated statistics and performance under them - i.e. section 4.1 and Figure 2
- No code for reproducibility

---

> ### Author Response · Authors · 2024-12-02
> **Response**
>
> We thank the reviewer for their feedback!
>
> > I would have appreciated a further study and insights on the aggregated statistics and performance under them - i.e. section 4.1 and Figure 2
>
> We expand on this and will continue to expand more.
> Added text: "The positive relationships between smiling and mouth_slightly_open and high_cheekbones seems intuitive while the positive relationship between pointy_nose and arched_eyebrows seems unintuitive and likely due to a spurious correlation. This relationship is explored on a single example in §4.2."
>
> > No code for reproducibility
>
> The code for the project has already been released but was not shared for this review for anonymity. A relevant file for computing CF alignment was uploaded as the supplementary material now. The link to the full source code repository which contains complete working colab notebook demos will be included, it is too difficult to anonymize the demos for this review.

---

### Review · Reviewer_93D4 · 2024-11-19

**Summary Of Contributions:**

The authors use an adapted version of the latent shift technique from Cohen 2021 to asses the impact of spurious features on classifier performance. In order to do this, the authors propose a method which uses an encoder/decoder model and a classifier which predicts a base feature. To the form counterfactual sample, they adjust the latent representation by the gradient of the classifier composed with the decoder, and then feeding the adjusted representation back into the decoder. They use this to form a metric which measures how much a new classifier of a different features uses the base feature by seeing the new classifiers prediction changes on the counterfactual sample.

The authors then give an experimental evaluation of their method. First they look at aggregate statistics for their metric over a dataset, demonstrating that their metric can go further than correlational based methods. They then examine specific examples, comparing against saliency maps. Finally, they do a series of experiments on group distributional robustness tasks, showing that state of the art models in the field are less susceptible to spurious correlations as measured by their metric.

**Audience:**

Yes

**Claims And Evidence:**

No

**Requested Changes:**

- Can the authors comment on what issues would occur if $f_b$ used spurious features?
- Can the authors comment on if the size of the metric can only be assessed relatively?
- Can the authors include comparisons with other counterfactual explanation methods? Comparing against Wachter 2018 would be good for a simple baseline but ideally also against some of the counterfactual explanation methods listed in the related work.
- Improve the clarity of definitions in section 3.

**Strengths And Weaknesses:**

Strengths:
- The method itself is easy to understand.
- As mentioned in the related work, the idea of modifying latent representations to produce counterfactual samples already has a basis in the literature.
- The results on the CelebA dataset are intuitively easy to understand.

Weaknesses:
- It would seem that if the base classifier uses spurious correlations, you may also end up creating counterfactual samples which change other features of the sample. Put another way, given I find a high relative change metric for a classifier $f_1$ and $f_b$, why can I conclude this is a problem with spurious correlations in $f_1$ alone and not $f_b$ also? To draw a conclusion like this, I would have to know $f_b$ does not rely on spurious correlations, which if assessed by relative change metric would start to feel like a circular argument.
- I find the experimental comparisons to be slightly lacking in that whilst the authors compare against saliency maps and correlational metrics, they do not compare against other counterfactual explanation methods. The relative change metric could be defined for any counterfactual sample and it would be informative to see how their method compares against such alternatives.
- It seems unclear to me how you would asses what a large or small value for the relative change metric is apart from relatively to other evaluations of the same metric. This can be seen in the fact that between experiments there is a large difference in the magnitude of the relative change metric.
- The definitions in section 3 could be clearer. The relative change metric should be brought out as an actual definition given its importance to the paper. Also at later points in the paper the authors seem to refer to the same metric as counterfactual alignment. If this is in fact the same this should be included in the definition.

---

> ### Author Response · Authors · 2024-12-03
> **Response**
>
> We thank the reviewer for their feedback!
>
> > Can the authors comment on what issues would occur if $f_b$ used spurious features?
>
> The question you may be asking is if one of the downstream classifiers (e.g. $f_1$) used spurious features? If this is the case then their output would change if the spurious features they used were modulated causing what could be considered a false positive. This is a limitation of this work. For this reason in this work we reference the classifier target names (e.g. mouth_slightly_open) instead of the concept that it is assumed to represent. One solution is to utilize multiple classifiers that are trained to predict the same concept using multiple training datasets so on average they predict the correct concept. We will expand this discussion in the limitations section.
>
> > Can the authors comment on if the size of the metric can only be assessed relatively?
>
> The relative change metric value can be utilized absolutely, a value of 1 indicates a very high alignment, 0 indicates no alignment/independence, and -1 indicates a strong inverse alignment. Beyond this the values are likely only relatively comparable as the classifier outputs can respond at different levels to spurious feature changes. Perhaps there is deeper calibration that can be explored in later work. We will add this to the paper.
>
> > Can the authors include comparisons with other counterfactual explanation methods? Comparing against Wachter 2018 would be good for a simple baseline but ideally also against some of the counterfactual explanation methods listed in the related work.
>
> The goal of this paper was to demonstrate this approach works, not claiming superiority. It would be nice to have this comparison as well. We will try to implement the Wachter baseline to see if it can provide similar results.
>
> > Improve the clarity of definitions in section 3.
>
> We added this text and also explicitly titled the equation for relative change.
>
> "The CF alignment approach tests if a base classifier $f_b$ utilizes features that are also used by a downstream classifier (e.g. $f_1$). If they do, the generated CF would have features modulated which would cause the other classifier (e.g. $f_1$) to have a different output. This alignment can then be measured quantitatively by comparing how aligned the changes in prediction are."

---

> > ### Author Response · Authors · 2024-12-13
> > **Response**
> >
> > To address another reviewers comments we expanded even more on the limitations when the downstream classifiers contain their own spurious correlations. The following text was added to the paper to discuss this issue:
> >
> > "Spurious alignment: This situation can occur if the downstream classifier ($f_1$) also has the same spurious features as the base classifier ($f_b$). If this is the case, then both of their output would change if the spurious features were modulated causing these classifiers to appear aligned. For example, if a base classifier predicts \emph{cow} but spuriously uses the presence of a beach to produce a low prediction. A CF generated for this classifier may change the background of a farm to a beach. If we check the alignment with a classifier that predicts \emph{chicken} (but this chicken classifier also reduces its prediction if the image contains a beach) then these classifiers would appear aligned and the relative change would be positive. These classifiers are in fact aligned based on feature usage but just because they just use the same spurious features. This may be similar to a false positive but it isn't false, the classifiers are aligned.
> >
> > For this reason, in this work we reference the classifier target names (e.g. \emph{mouth\_slightly\_open}) instead of the concept that it is assumed to represent. One solution to avoid these edge cases is to utilize multiple classifiers that are trained to predict the same concept using multiple training datasets so on average they predict the correct concept.
> >
> > False negative alignment: This situation can occur if the downstream classifier ($f_1$) predicts using features that are "opposite" of the base classifier ($f_b$). For example "opposite" can mean the base classifier ($f_b$) predicts \emph{cow} using only the presence of absence of a beach while the downstream classifier ($f_1$) predicts \emph{beach} using only the presence of absence of a cow. The CF for the base classifier changes the background from a beach to something else but this doesn't change the prediction of the downstream classifier ($f_1$) so there is no alignment, the relative change would be 0. This case could be considered a false negative because we didn't find alignment with a \emph{beach} classifier but the \emph{beach} classifier was flawed. To address an issue like this we could simply inspect the CFs that are being generated with domain experts as well as vary the downstream classifier's training data."

---

> ### Author Response · Authors · 2024-12-17
> **Response**
>
> > Can the authors include comparisons with other counterfactual explanation methods? Comparing against Wachter 2018 would be good for a simple baseline but ideally also against some of the counterfactual explanation methods listed in the related work.
>
> In the new revision the appendix section was added "Comparison with other CF generation methods" where we compare to Wachter 2018. While this experiment is limited, if it doesn't work for the known biased classifier experiment it likely won't work for the other experiments. There is much more that could be done here but time is limited during this rebuttal period. We hope this experiment is enough to make this work complete.

---

> > ### Comment · Reviewer_zg5u · 2024-12-17
> >
> > I am a different reviewer, and I would just like to add that I believe that *''Comparing against Wachter 2018''* is not particularly meaningful, considering that the method used in the manuscript is essentially an instance of the method by Wachter 2018. I am willing to stand up for the authors here.

---

> > > ### Comment · Reviewer_93D4 · 2024-12-19
> > >
> > > I thank the authors for their detailed response, especially the added paragraph on the limitations section. I would also like to thank Reviewer zg5u for their input on the comparison to Wachter 2018, I had not fully appreciated the direct comparison and am grateful that the reviewers took the steps to accommodate my suggestion anyway.
> > >
> > > Overall the authors revisions have satisfied all my concerns and so I will recommend accept.

---

### Decision · Action_Editor_t6Vt · 2024-12-20

**Recommendation:** Accept as is

**Comment:**

The authors adjusted the experiments as discussed and revised the text to address the comments related to the omissions and discrepancies between the claims and the evidence, with reviewers noting that the comments have been sufficiently addressed.  All reviewers recommend acceptance and justify it by stating that 'the methodology is novel and interesting', 'being simple to implement and understand', 'sufficient experimental results', 'elegant, rigorous methodology'

**Audience:**

All reviewers suggest that the paper and the approach has audience as the identification of spurious correlations is an important problem.

**Claims And Evidence:**

The authors propose the counterfactual alignment method for spurious correlation detection and quantification.  The authors use two classifiers: the base one and the downstream one. They compare the relative change of prediction of the downstream classifier given the counterfactual image generated with respect to the base classifier.

After a number of revisions thoroughly discussed with the reviewers, all reviewers suggest that the paper meets the criteria of claims and evidence.